# Geology controls the distribution of a seed-eating bird: Feeding-tree selection by the glossy black-cockatoo *Calyptorhynchus lathami*

**Gabriel M. Crowley** *

Department of Geography, Environment and Population, School of Social Sciences, Faculty of Arts, Business, Law and Economics, The University of Adelaide, Adelaide, South Australia, Australia

* gay.crowley@adelaide.edu.au

**Data Availability Statement:** All data are in the manuscript and/or supporting information.

## Abstract

Despite seed production being nutrient-limited, the influence of nutrient pathways on granivore distributions is unclear. This article examines the influence of geology and soil on the distribution of glossy black-cockatoos (*Calyptorhynchus lathami*), which feed almost exclusively on the kernels of casuarinas (*Allocasuarina* spp. and *Casuarina* spp.), and are selective about the trees in which they feed. To clarify the basis of this selection, Food Value (a measure of dry matter intake rate) and kernel nutrient content were compared between feeding and non-feeding trees of drooping sheoak (*A. verticillata*). Random forest modelling was then used to examine the influence of geology and soil chemistry on Food Value. Finally, logistic generalised additive modelling was used to examine the influence of geology on cockatoo feeding records across the range of black sheoak (*A. littoralis*) and forest oak (*A. torulosa*), drawing on a statewide dataset. Food Value–but not kernel nutrient concentrations–influenced feeding tree selection. Soils under drooping sheoak were nutritionally poor, with low nitrogen and phosphorus (despite high concentrations of these nutrients in the kernels), and characterised by two principal components: SALINITY (dominated by exchangeable magnesium and sodium, electrical conductivity, and sulphur) and ACIDITY (pH, iron, and aluminium). Random forest modelling showed that Food Value was highest on sedimentary rocks, with a high ACIDITY score, less than 18 meq 100 g$^{-1}$ exchangeable calcium, and less than 4% soil organic carbon. The odds of cockatoos selecting casuarinas as feedings tree were three times higher on non-calcareous sedimentary rocks than on other rock types. Non-calcareous sedimentary rocks produce low-fertility, acid soils, which promote nitrogen-fixation by *Frankia*. I therefore conclude that glossy black-cockatoo distribution is controlled by the casuarina's symbiotic relationship with *Frankia*, which is ultimately controlled by geology; and that similar relationships may be responsible for the prevalence of several other species on low-fertility and/or acid soils.

## Introduction

All biological systems are nutrient-dependent [1]. In terrestrial ecosystems, most nutrients are ultimately derived from the parent rocks [2], although nitrogen (N) may also be derived from

**Funding:** Funding for sample collection in 1996-1997 was provided by the Threatened Species and Communities Unit of Environment Australia, with additional funding from the South Australian Department of Environment and Heritage and the Glossy Black Rescue Fund of the National Parks Foundation, South Australia. The funders had no role in study design, data collection and analysis, decision to publish, or preparation of the manuscript.

**Competing interests:** The author has declared that no competing interests exist.

the atmosphere through nitrogen-fixing symbiotic microorganisms [3], and organic carbon (C) from plants and soil biota [4]. The most fertile soils are found on clay-rich rocks, such as basalt, limestone and fine-grained sedimentary rocks [5]. In addition, weathering (the product of climate, topography and biological activity) initially makes nutrients available, and–over time–leaches them from the soil profile. By changing the soil's chemistry, texture, and pH, weathering affects the ability of the plant to take up nutrients, including by influencing the efficacy of N-fixation [3]. Hence, plant productivity tends to be limited by N on young soils, and by phosphorus (P) on ancient soils [6]. These limitations flow on to the animal populations that can be supported by the vegetation, which may be additionally limited by calcium (Ca) needed for bone and egg formation [7–9]. Hence, rocks–and the soils derived from them–have the capacity to control the distribution and productivity of plants and, thereby, of the animals that are dependent upon their seeds, fruits, flowers and leaves [10].

Such nutrient pathways have been best demonstrated in the herbivorous red grouse (*Lagopus lagopus scotica*) in Scotland [7, 11] and the invertebrate-feeding ovenbird (*Seiurus aurocapilla*) in Pennsylvania [8]. The grouse feed on a small number of plant species, and were found to be more abundant and have greater breeding success on basic rocks containing limestone, than on acidic granites. These differences were correlated to differences in the composition, abundance and N content of food plants on the two substrates. The ovenbird's abundance and clutch size was correlated to snail abundance, which in turn was highest on alkaline soils overlying limestone rocks.

Other studies examining the geologic or edaphic control of animal distributions have been more fragmentary, involving only part of the nutrient pathway. For example, geological control of animal abundance has been demonstrated in several species in Australia [12, 13], Africa [14], and North America [15]. In most of these cases, the assumption has been made that this relationship was mediated through soil fertility, although the production of toxic substances is also thought to have had a role [16], such as in the case of serpentine rocks [15]. Conversely, without exploration of the underlying geology, soil fertility and/or calcium content–either low or high–have been shown to favour the abundance of other animals in Africa [17], Australia [18, 19], and Britain [20]. While nutrient limitation may explain why some animals appear to be most abundant or have highest breeding output on rocks that produce soils of high fertility [12–14, 19], or high calcium content [8, 9, 18], it cannot explain why the reverse is true for other animals [9, 18, 19], or where an affinity with particular rock types does not appear to be mediated through nutrient status [21].

There appear to have been no studies of the influence of geologically-driven nutrient flows on seed-eaters. This is despite the fact that seed production is macronutrient-limited [22], and exerts bottom-up control on granivorous bird populations [23, 24]. Geological influence on granivore abundance is likely to be obscured by other bottom-up (e.g. climate, water-availability, vegetation structure, nest sites) and top-down (e.g. predation, disease) processes, disturbance (e.g. fire, flood), and competition from other species [25, 26], as well as the ability of species to switch between food sources in response to changes in seed availability [27, 28]. Its study will therefore be most successful where such complexities can be minimized.

The glossy black-cockatoo (*Calyptorhynchus lathami* Temminck, 1807) feeds almost exclusively on the kernels of nine species of casuarina (*Allocasuarina* spp. L.A.S. Johnson and *Casuarina* spp. L), which produce abundant seeds that are rarely taken by other co-occurring granivores [29]. Each cockatoo feeds on an average of 4–5 trees per day, taking an average of 61–78 cones per day in the non-breeding season, and 123–128 cones per day in the breeding season [30]. It systematically shreds each cone to extract the seeds one by one, neatly bisecting each in its bill to remove and ingest the single kernel [31]. The casuarinas gain little benefit from this exploitation, as the cockatoos usually extract and consume all viable seeds from each

cone they handle [31], though occasionally drop half-shredded cones that still contain seeds [30].

The cockatoos also require hollow-bearing eucalypts in which to nest [32] and daily access to water [33]. Their distribution is therefore confined to areas of the Australian mainland where feeding and nesting habitat coincide, and standing freshwater is available through the year, namely in subcoastal environments between South Australia and northeast Queensland [34, 35]. However, not all areas that meet these criteria contain cockatoos, and the patchy distribution of the species has long perplexed ornithologists. The species is long-lived [36], largely disease-free [37] and–before clearance for agriculture fragmented the habitat–experienced minimal predation pressure or competition for nest hollows [38]. Gaps in the expected distribution may therefore result from nutritional limitations. Not all casuarina cones produce enough kernel matter to sustain the cockatoos, and the birds will reject trees that provide a suboptimal dry matter intake rate [31, 39]. Recent work has also suggested that they may also discriminate on seed nutrient content [40]. As seed production and nutritional composition is influenced by nutrient uptake from the soil, and therefore by the parent rock [41], I propose that the distribution of glossy-black cockatoos is influenced by geology.

This article, therefore, examines how geology influences glossy black-cockatoo distribution through nutrient pathways. It describes three studies. Study 1 re-assessed which kernel characteristics control feeding tree selection by the cockatoos. Study 2 examined the influence of geology and soil on these controlling characteristics. Study 3 asked whether any geological influence identified in Study 2 operates at a broader scale. The article then explores the implications of the findings for both the management of this threatened species, as well as for understanding how nutrient flows might operate in other ecological systems.

## Methods

### The study areas

Studies 1 and 2 were undertaken within the range of the drooping sheoak (*A. verticillata* (Lam.) L.A.S. Johnson), from the Eyre Peninsula in South Australia to central-western New South Wales, including Kangaroo Island (34–38˚ S, 136-147˚ E; Fig 1). The area has a semi-arid to subhumid, temperate climate. Average monthly temperatures range from 5-16˚ C in July to August to 15-30˚ C in January to February [42]. Annual rainfall averages 400–800 mm. Rainfall is evenly distributed through the year in the northeast, and increases in seasonality to the southwest, where two-thirds falls in May to September.

Study 3 was undertaken within the range of black sheoak (*A. littoralis* (Salisb.) L.A.S. Johnson) and forest oak (*A. torulosa* (Aiton) L.A.S. Johnson) in subcoastal New South Wales (28-38˚ S, 148-154˚ E). The area has a subhumid to semi-arid, warm temperate to subtropical climate with no dry season, receiving between 700 mm and 2,000 mm annual average rainfall. Average monthly temperatures range from 5-15˚ C in July to August to 18-27˚ C in January to February [42].

### The species

Drooping sheoak is a shrub to small tree that forms monospecific stands of low woodland/tall shrubland on coastal sands or shallow soils over sandstone, limestone or basalt [43, 44]. It also occurs as an understorey species in open forest on acidic gradational soils over sandstone or metasandstone, frequently in association with sugar gum (*Eucalyptus cladocalyx* F. Muell.) or other eucalypts [43, 44]. It is fed on by glossy-black-cockatoos on Kangaroo Island and in the Riverina, New South Wales. Black sheoak (a shrub to small tree) grows mainly on poor sandy, skeletal or rocky soils [43, 44]. Forest oak (a medium-sized tree) is found on deeper, more

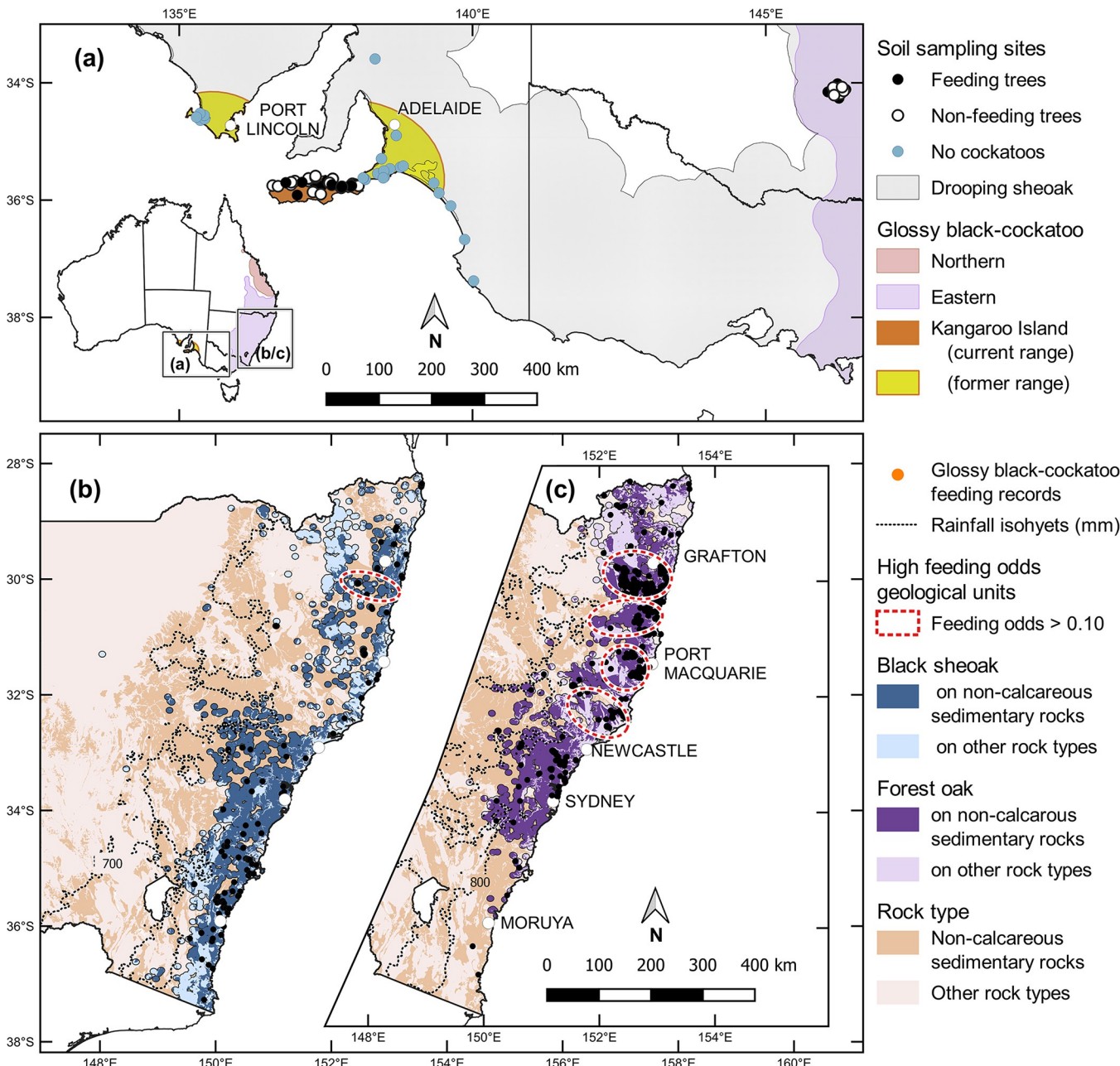

**Fig 1. Study areas showing the distribution of glossy black-cockatoo subspecies and the three casuarinas examined in this study.** (a) Studies 1 and 2, drooping sheoak. Study 3 (b) black sheoak, and (c) forest oak.

fertile soils that overlay basalts and a range of igneous and sedimentary rock types [43, 44]. Both grow in the understorey of eucalypt-dominated open-forests and tall open forests in eastern Australia, although black sheoak also forms monospecific stands on rocky granitic outcrops [43]. Both are fed on by glossy-black-cockatoos along the eastern Australian seaboard [29].

Casuarinas produce woody cones, each with up to ~100 seeds [31]. The seed has a single kernel and a winged seed coat (samara), which aids dispersal on cone maturity (in *Casuarina*), or after fire or stem death (in *Allocasuarina*) [45], although the majority of seeds fall below the

canopy [46, 47]. The seeds have protein-rich kernels [ca. 44% protein; 31], which require high concentrations of N, P, Ca, sulphur (S), magnesium (Mg), potassium (K), and iron (Fe) [22]. In most plants, these nutrients are derived from the soil, often with the aid of symbiotic organisms [2, 6]. Casuarinas belong to a suite of non-leguminous plants that derive N from the atmosphere through a symbiotic relationship with the N-fixing actinorhizal bacterium *Frankia* Brunckhorst 1885 [48], which is influenced by soil chemistry [3]. Their acquisition of other nutrients is enhanced through endomycorrhizal and/or ectomycorrhizal associations and cluster root formations [49].

The glossy black-cockatoo is a small (400–520 g) member of Australia's endemic Calyptorhynchinae cockatoos [32, 50]. The males are black with red windows in the tail. The females are similar, but each has a unique combination of yellow patches on the face and neck, and the red windows are interspersed with black bars. There are three subspecies [recently amalgamated into a single taxon; 50], which occur through much of subhumid to semi-arid eastern Australia (Fig 1). All are classified as threatened, largely because of habitat loss, which was exacerbated by extensive bushfires in 2019/20 (S1 Table). In these fires, the nominate subspecies lost between one-quarter and one-third of its habitat [51], and the Kangaroo Island subspecies lost 54% of its feeding habitat [52]. The species exhibits mate fidelity and nests in hollows. Each pair lays a clutch of one egg, raising only one fledgling per year [32]. The female alone feeds the single nestling, and is, in turn, fed by the male when she is incubating or brooding [32].

Calyptorhynchinae cockatoos have strong bills that allow them to extract seed from woody cones. While most species are arboreal foragers, and several eat casuarina seeds in small quantities [35], only the glossy black-cockatoo has specialised to the extent of feeding almost exclusively on a single food source–the kernels of casuarinas [29]. This specialisation evolved around 7 million years ago [50, 53] as aridity developed across the Australian continent and casuarina woodlands expanded [54]. It was possible because casuarina kernels meet the full nutritional needs of the cockatoos [31], to the extent that the birds spend only 26% of their time feeding in the non-breeding season and 36% in the breeding season [30].

When feeding, a cockatoo selects a cone from the tree, and bites into it to remove a single seed, which it bisects to remove and ingest the kernel [31]. It will repeat this process until all the kernels are eaten. Usually limited to feeding on one or two species in any area [29], the cockatoos are also selective about which trees and stands they use [55]. They forage to maximize dry matter intake rate [31], selecting feeding trees in which a high percentage of seeds contain sound kernels (Seed Fill) and in which the weight of an individual kernel is high relative to the cone's weight (Kernel Ratio). These two variables combine to produce the cone's Food Value to the cockatoo [ratio of total kernel weight to cone weight; 31]. Clout's Index (ratio of total seed weight to cone weight) provides a close approximation of Food Value [31], and has been widely adopted because of its ease of measurement [e.g. 40, 56, 57]. Selection of feeding tree is also influenced by tree size [which may be a surrogate for the size of the cone crop; 56], but not by the addition of NPK fertilisers to the soil [58], nor by kernel concentrations of elements, protein, lipid or energy [31]. Even though the cockatoos do not eat whole seed, higher concentrations of several elements (N, Fe, aluminium (Al), Copper (Cu), Boron (B), Silicon and Strontium) were found in the whole seed of feeding trees than in those of non-feeding trees; with the suggestion that tree selection may also be based on individual nutrient requirements [40]. Whether this is the case needs to be clarified.

### Studies 1 & 2—Data collection

Study 1 (reassessing the influence of seed variables on tree selection by the cockatoos) and study 2 (examining the influence of geology and soils on these controlling variable(s)) used

seed samples from a dataset that had been produced for an earlier publication that examined drooping sheoak tree selection by glossy black-cockatoos on Kangaroo Island [31]. As well as the 76 samples used in the earlier publication, this dataset also included additional samples from Kangaroo Island, as well as samples from the Riverina and mainland South Australia. All samples were collected between January 1996 and August 1997. In total, 551 cone samples had been collected from 330 drooping sheoak trees. 170 samples from 143 trees contained data that were relevant to the current studies (S1 Dataset). Seed Fill, Kernel Ratio, Food Value, and seed and kernel chemistry, geology and soil chemistry were available for various subsets of samples, as described below. Trees were allocated to a cockatoo status class: "feeding" or "non-feeding" within active foraging areas (based on the presence or absence of chewed cone remnants below the tree); and "absent" for trees in areas with no cockatoo records for at least 20 years [59].

Each cone sample was a minimum of 10 mature (red to brown) cones from the current year's crop and was dried at 60°C to constant weight, following Clout [39], a procedure that has been adopted by subsequent authors [31, 57], [though sometimes with smaller sample sizes; 40]. In each of these publications, this protocol has been shown to successfully distinguish between feeding and non-feeding trees. Dry weights were calculated for each cone sample before seed extraction, cone samples after seed extraction, seed samples, and kernels extracted from a randomly-selected subsample of 100 seeds. Kernel weight was averaged from the sound kernels in these seeds. Seed Fill was calculated for all batches of dissected seeds. Kernel Ratio was calculated from individual kernel weight/individual cone weight; and Food Value from the weight of all kernels in a sample /weight of all cones in a sample. Food Value had only been calculated for samples collected after May 1997, and soils from before June 1997. N content was measured for 43 kernel and whole seed samples with at least 10 kernels, and concentrations of B, Ca, Cu, Fe, K, Mg, Mn, Na, P, S and Zn were determined for 45 kernel and whole seed samples using the methods described in Crowley and Garnett [31].

For Kangaroo Island sites, parent rock was determined using the 1:50,000 Geological Map Series (A. P. Belperio, South Australian Department of Mines and Energy, unpublished data June 1996). Mainland rock types were assessed from the 1:1,000,000 Surface Geology map [60]. Rock types were grouped into unconsolidated sand or alluvium, igneous rock, and sedimentary rock. Soil samples of at least 500 g were collected from the top 5 cm in at least three locations beneath the tree canopy after the litter layer had been removed after Yadav [61]. Soils were dried to constant weight at 60°C; sieved to < 2 mm, and then analysed for nutrients by CSBP, Perth, Australia, as follows. Soil texture was assessed using manual manipulation [62]. The following characteristics were measured using the methods of Rayment and Lyons [63]: pHc (pH using CaCl$_2$ extraction, method 4B4), pHw (pH using water extraction, method 4A1), available P (Colwell, method 9B), available K (Colwell, method 18A1), N as nitrate and ammonium (method 7C2b), organic C (method 6A1), S (KCl 40, method 10D1), Fe (Tamms reagent, method 13A1) and electrical conductivity (EC, water extraction, method 3A1). Al (CaCl$_2$ extraction) and exchangeable Ca, K, Mg and sodium (Na) were measured using the methods of Gillman and Sumpter [64].

## Studies 1 & 2—Data analysis

All statistical analyses were undertaken in R [65]. To avoid the impact of non-independent samples [66], where duplicate samples were available from an individual tree, a single sample was randomly selected for inclusion in the analysis.

As Food Value was unavailable for most trees with soil samples, Food Value (FV) was predicted from the best-fit model explaining Seed Fill (SF) and Kernel Ratio (KR) of 77 trees for

which all three variables were available. To minimise the influence of outliers, robust regression from the R package robustbase [which down-weights cases with large residuals based on a bi-square redescending score function; 67] was used with combinations of untransformed and log-transformed dependent variables to identify the best-fit model. As Seed Fill was not significantly correlated with Kernel Ratio (r = 0.112, P = 0.334) and log-transformation reduced the correlation between variables in all cases (S2 Table), all combinations of these two variables provided valid models. Based on the lowest robust residual standard error, the best-fit model (adjusted $R^2$ = 0.685; S3 Table, S1 Fig) was selected as:

$$FV = 40.623 \times SF + 11.629 \times \log(KR) + 9.7193$$

This relationship was used to predict Food Value for all samples for which both Seed Fill and Kernel Ratio were available.

For all other analyses, the distribution of data was assessed with the following tools from the R package rstatix [68] to determine the appropriate tests to use and/or whether transformation was required to improve normality. The Shapiro test was used to assess normality, an identify outliers and extreme outliers, and–where appropriate–the Levene test was used to assess homogeneity of variance. To assess whether dry matter intake rate was the most important variable controlling tree selection by the cockatoos (Study 1), predicted Food Value was compared between cockatoo status classes using a Kruskal Wallis test with Wilcoxon test pairwise comparisons (as the distributions of each class were markedly different); and logistic regression was used to further investigate the relationship between tree selection and Food Value established by Crowley and Garnett [31]. To assess the role of individual nutrients in tree selection, the elemental composition of kernel samples and the whole seed batches from which they were extracted were compared using paired t-tests, if the assumptions of normality, equal variance and extreme outliers were met. Otherwise, pairwise Wilcoxon tests were used. Similarly, t-tests or Wilcoxon tests were used to assess whether the elemental composition of kernels and of whole seed differed between feeding and non-feeding trees. Significant P-values (P < 0.05) for each collection of tests are reported both before and after correction for false discovery rate using Benjamini-Hochberg (BH) adjustment [69].

Informed by the above analysis, the influence of geology and soils on predicted Food Value was examined using random forest modelling [70]. Random forest is a robust machine learning technique that compares multiple decision trees to describe the relative importance of intercorrelated variables to an outcome, and is increasingly being used in the examination of complex ecological systems [71]. As multicollinearity between soil variables can make it difficult to isolate the influence of individual variables [72], Principal Components Analysis (PCA) was used to combine intercorrelated numeric variables into independent variables, which were also included in the random forest modelling. To reduce the influence of different measurement scales and extreme values, and to maximise variance, soil variables were first transformed, centred and scaled (S2 Fig). Al (with multiple zero values) was quarter-root transformed, and all other nutrients were $\log_e$-transformed. pH measures were scaled but not transformed. After this treatment, Pearson's correlation confirmed collinearity between several of the variables (S3 Fig), and so a PCA was undertaken using the R package psych [73] with a covariance matrix and oblimin rotation. A scree plot depicting eigenvalues, Parallel Analysis, Optimal Coordinates and Acceleration Factor [74] was used to determine the number of components to use in the PCA. For visual display, loadings were calculated for each contributing variable, and vectors for predicted Food Value and three constructed soil variables (ammonium/nitrate, Organic C/N, and total N) were fit to the principal components using the envfit

function in the R package vegan [75]. Scores for each principal component were extracted for each sample to use in the random forest modelling.

Predicted Food Value data for samples with associated rock and soil samples were moderately negatively-skewed (Shapiro W = 0.933, P < 0.001, four outliers and no extreme outliers; S4 Fig). Therefore they were square-root transformed before use in random forest modelling ($\sqrt{(max(cFV) + 1)} - cFV$), where pFV = predicted Food Value), which improved normality (Shapiro W = 0.986, P = 0.508, no outliers). These random forest models were run using the R package randomForest [76] using cross validation 10 times with three repeats. The models were tuned to optimise the number of variables to be randomly sampled at each split in each tree using the train function in the R package caret [77], and the number of decision trees to be compared to ensure that the models stabilised. An initial model was run that included rock type, soil texture, the original, untransformed data for 15 measured soil variables, three constructed soil variables, and scores for each of the principal components as independent variables. A final parsimonious model was produced using the forward stepwise elimination process in the R package VSURF [78], in which extraneous variables were excluded if the standard deviation of their Variable Importance exceeded the minimum value predicted by a pruned classification and regression tree fitted to the standard deviation curve, or they made a negligible or negative contribution to the Out of Bag error.

In response to the random forest model results, non-calcareous sedimentary rocks and other rock types sampled were compared in relation to scores for the first PCA component, Food Value and kernel concentrations of the 12 available elements using t-tests (where the assumptions of normality, equality of variance and extreme outliers could be met) or Wilcoxon tests. Significant P-values (P < 0.05) are reported both before and after BH correction for false discovery rate. Sampling bias precluded using this dataset for assessing the influence of rock type on cockatoo status or feeding tree selection.

## Study 3—Data collection

Study 3 examined the influence of geology on glossy black-cockatoo feeding records within the range of black sheoak and forest oak in New South Wales (NSW), informed by the results of Study 2. All available records of black sheoak (12,030 records) and forest oak (11,454 records) were downloaded from NSW BioNet on 18 April 2020 [79]. In addition, all glossy black-cockatoo feeding records (6,543 records) were downloaded on 12 April 2020. Both datasets were filtered using information provided in the original datasets to exclude records with a spatial accuracy of more than 2 km, as well as those missing key spatial or temporal collection data, or that were of cultivated plants, and retaining only those records from within New South Wales, as determined using Quantum GIS [QGIS; 80] (S4 Table). Feeding records were further filtered to include only those records in which either feeding had been observed in black sheoak or forest oak, or the distinctive crushed cones [81] had been recorded under one of these *Allocasuarina* species. Each record was allocated to a 1 ha (100 m x 100 m) UTM grid cell (EPSG: 32755 and 32756) in QGIS. To reduce biases arising from variable sampling intensity and possible duplicate records, all records for a single grid cell were amalgamated to indicate whether each casuarina species was present, and–if so–whether it had been fed on by the cockatoos. This resulted in a dataset of 16,922 one-hectare grid cells containing one or both *Allocasuarina* species, including 938 cells containing 6,543 records of glossy black-cockatoos feeding on these species (S2 Dataset). As the glossy black-cockatoo is a threatened species in NSW, location of all grid cells was dithered by setting the minimum easting and northing values to zero, and adding a random distance of between 100 m and 1,000 m to each resultant easting and northing.

Before this spatial dithering, each grid cell was allocated a geological map unit by intersecting its midpoint with a 1:1,000,000 geological map [60]. Informed by the results of Study 2, map units were classified according to the dominant rock type as either sedimentary or other, and calcareous or other (S5 Table). Sedimentary rocks included sandstone, mudstone, claystone, siltstone, metasediment and conglomerate. Non-calcareous rocks were rock units in which limestone, marble and calcarenite were absent or described as minor or rare components.

### Study 3—Data analysis

Binomial generalised additive modelling in the R package mgcv [82] was used to assess the influence of the fixed effects of rock type and *Allocasuarina* species on the probability of a grid cell containing feeding records, accounting for spatial location as a random effect. Two models were run, one including and one excluding interactions between species and rock type. The preferred model was selected based on minimum residual deviance [83], followed by Akaike Information Criterion [AIC; 84], and lastly by Bayesian Information Criterion [BIC; 85]. Highly productive geological units were identified for each *Allocasuarina* species based on the presence of at least 50 grid cells containing *Allocasuarina* records and an odds ratio of feeding cells to non-feeding cells greater than 0.1.

## Results

### Predicted Food Value and tree selection

Predicted Food Value of drooping sheoak cone samples ranged from 2.8 to 39.8 mg g$^{-1}$ (Fig 2A) and differed significantly between cockatoo status groups (Kruskal-Wallis $\chi^2$ (d.f. = 2) = 42.4, P < 0.0001; Fig 2B). Wilcoxon test for pairwise comparisons indicated that predicted

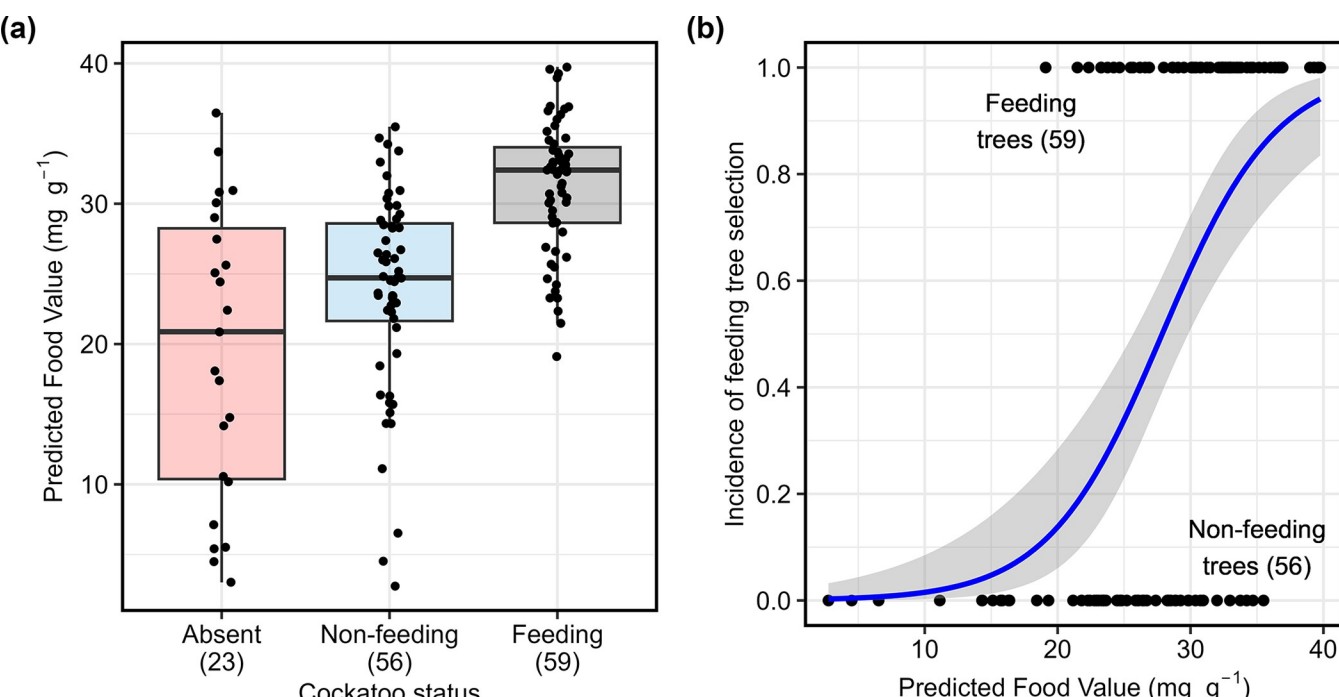

**Fig 2. Comparison of predicted Food Value of drooping sheoak in relation to glossy black-cockatoo distribution and feeding tree selection.** (a) Predicted Food Value in relation to distribution of, and tree selection by, glossy black-cockatoos. (b) Glossy black-cockatoo tree selection within the range of the cockatoos in relation to Predicted Food Value. Sample sizes are shown in brackets.

Food Value of feeding trees was significantly higher than that of either non-feeding trees or trees from areas without cockatoos (P < 0.0001), but not between non-feeding trees and trees from areas without cockatoos (P = 0.120). Within the current glossy black-cockatoo's distribution, the influence of predicted Food Value on tree selection (mean ± s.e.m: feeding trees (n = 59), 31.3 ± 0.62 mg g$^{-1}$; non-feeding trees (n = 56), 23.8 ± 0.97 mg g$^{-1}$) was confirmed by logistic modelling. Incidence of tree selection (iTS) was significantly influenced by predicted Food Value (pFV), with the relationship (Fig 2B):

$$iTS = 0.2331 \times pFV - 6.495$$

This model significantly reduced the residual deviance over the null model from 159.4 to 121.1, with a degrees of freedom reduction from 114 to 113 (P < 0.0001; S6 Table).

## Seed nutrients

N was the most abundant element in the kernels of drooping sheoak (mean ± s.e.m. = 7.18 ± 0.06%), and had the lowest coefficient of variance (c.v. = 5.9%; S7 Table). Based on a conversion factor of 6.25 [86], this equates to a crude protein content of 44.9 ± 0.6%. The next most abundant elements were P (1.40 ± 0.04%, c.v. = 19%) and K (1.08 ± 0.02%, c.v. = 12%). S was the fourth most abundant element (0.482 ± 0.005%) and had the second lowest c.v. (7.3%). Kernel Ca concentration was 0.29 ± 0.11%, with a c.v. of 25%. Kernels of sound seed contained 94% of the N, 99% of the P, 92% of the S, 91% of the Mg, 86% of the Cu, 60% of the B, 56% of the Ca, and 28% of the Fe.

Paired t-tests and Wilcoxon tests (both before and after P-value adjustment for false-discovery rate) indicated that kernels had significantly higher concentrations of P, N, S, Mg, K and Mn (P < 0.0001, with ratios between 1.6:1 and 2.1:1) and Zn (P < 0.001, with a ratio of 1.15:1) than the whole seed batches from which they were extracted, and were significantly lower in Ca, Na, Fe (P < 0.0001, with ratios between 0.25:1 and 0.45:1) and Cu (P < 0.001, with a ratio of 0.83:1; S5 Fig). Only B concentrations did not differ significantly between kernels and whole seed (P = 0.642).

No significant differences were found between feeding and non-feeding trees in the kernel concentration of any element, either before or after P-value adjustment (P > 0.05; S6 Fig). Kernel concentrations in both feeding and non-feeding trees were approximately 7% for N, 1.4% for P, 1.1% for K, 0.5% for S and Mg, and 0.3% for Ca. Significant differences were found in the whole seed batches, with t-tests showing that feeding tree had lower Cu (P = 0.003, BH-adjusted P = 0.035; S7 Fig) and Zn (P = 0.006, BH-adjusted P = 0.035) than were found in non-feeding trees. Whole seed concentrations of N (P = 0.019, BH-adjusted P = 0.074) and Mn (P = 0.049, BH-adjusted P = 0.146) were significantly higher in feeding trees before, but not after, P-value correction.

## Geology and soils

Drooping sheoak was mostly found on sedimentary rocks, but also occurred on igneous rocks, laterite, and unconsolidated sands and alluvium (S8 Fig). The majority of soils were sandy loam, but the trees were also found on loam, clay-loam and clay. Soil chemistry was comparable to that from other soils sampled under drooping sheoak on Kangaroo Island [58]. However, in comparison to many other Australian soils [87, 88], they were typically more acidic (pHw, median: 6; interquartile range: 5.7–6.4). They were also rich in Fe (765 mg kg$^{-1}$; 490-1,170 mg kg$^{-1}$), with N dominated by ammonium (ammonium N/nitrate N: 3; 1.3-6.0). Their levels of organic C (3.6%; 3.0–4.7%) were high for soils collected under casuarinas. Total N (14

mg kg$^{-1}$; 9.8-25.3 mg kg$^{-1}$) was two orders of magnitude lower–and hence C to N ratios were two orders of magnitude higher (2,272; 1,540–3,244)–than found in most Australian soils [87, 89]. Available P (6 mg kg$^{-1}$; 3.0-11.0 mg kg$^{-1}$) was also at the low end of the scale [87], but available K (387 mg kg$^{-1}$; 240-497 mg kg$^{-1}$) was relatively high [88]. Ca typically dominated the exchangeable cations (57.0%; 48.7-65.3%), followed by Mg (26.0%; 21.1-30.1%), with Na (8.5%; 4.5-9.7%) and K (6.2%; 4.0–9.7%) having similarly low levels. Distributions of most nutrients were skewed, with maximum values several times the median. Nearly half the samples had no detectable Al, and one-quarter had values of more than 16 mg kg$^{-1}$. Most soils were of low salinity (EC range: 0.1-0.2 dS m$^{-1}$), but five samples exceeded 1 dS m$^{-1}$, and these also had extreme S concentrations (range: 124-314 mg kg$^{-1}$) and exchangeable Na concentrations exceeding 6 meq 100 g$^{-1}$.

Eigenvalues indicated that most variation in numerical soil values would be captured by three principal components (S9 Fig). However, Parallel Analysis, Optimal Coordinates and Acceleration Factor all indicated that two components would be sufficient. Therefore, two components were used in the PCA. PCA with two factors explained 68% of the variance in numerical soil variables. The first component (SALINITY; eigenvalue: 5.77) was dominated by exchangeable Mg and Na, EC and S (all with positive loadings), and explained 38% of the variance (Fig 3A, S8 Table). The second component (ACIDITY; eigenvalue: 4.42), was dominated

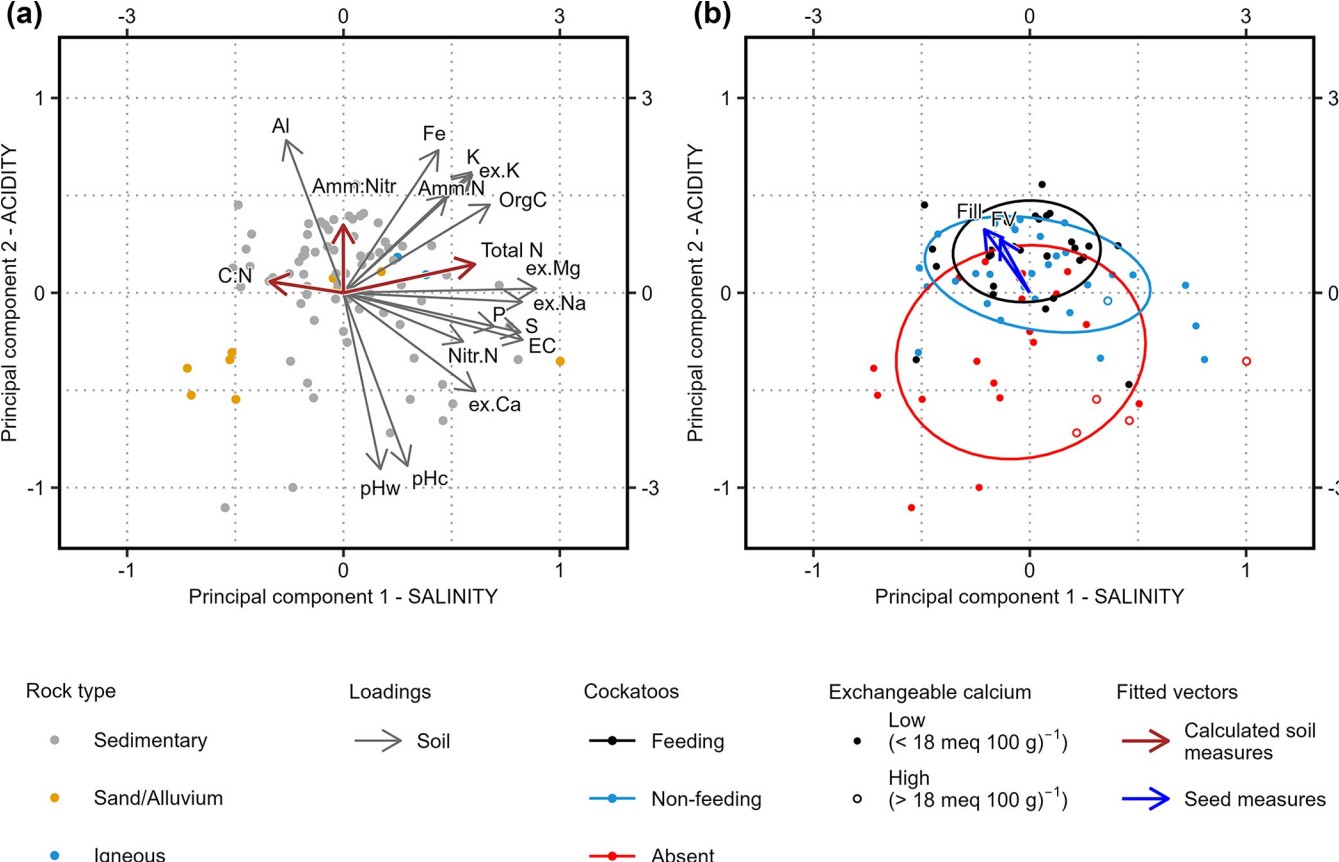

**Fig 3. Plot of Principal Components Analysis (PCA) depicting the variation in numerical soil variables beneath drooping sheoak.** (a) Loadings for soil variables and fitted vectors for constructed soil variables. (b) Fitted vectors for seed variables (displaying 75% confidence ellipses for each cockatoo class). PCA loadings and fitted vectors are mapped against the left and lower axes. PCA scores of individual sites are mapped against the right and upper axes. Abbreviations (other than for chemical elements): Amm, Ammonium; EC, Electrical Conductivity; ex, exchangeable; Fill, Seed Fill; FV, predicted Food Value; OrgC, organic C; Nitr, nitrate; pHw, pH (water extraction method); pHc, pH (CaCl$_2$ extraction method).

by pHw and pHc (with negative loadings) and Al and Fe (positive loadings), and explained 29% of the variance. Total N was moderately correlated to SALINITY ($R^2 = 0.389$, $P < 0.001$; S9 Table); ammonium/nitrate was weakly correlated to ACIDITY ($R^2 = 0.121$, $P = 0.016$), and C/N was weakly negatively correlated to SALINITY ($R^2 = 0.119$, $P = 0.008$). Seed Fill ($R^2 = 0.149$, $P = 0.002$) was significantly correlated to the principal components, with a stronger relationship to ACIDITY (0.840) than to SALINITY (-0.543; Fig 3B). The same was true for predicted Food Value ($R^2 = 0.093$, $P = 0.017$, ACIDITY score: 0.895, SALINITY score: -0.445); but Kernel Ratio was not significantly related to either component ($P > 0.05$). Feeding trees were most prevalent on soils with positive ACIDITY scores and neutral SALINITY scores.

## Influence of geology and soil on Food Value

The initial random forest model explaining predicted Food Value used rock type, soil texture, all 18 measured and calculated numerical soil variables and both soil principal components as explanatory variables (with 500 trees and 21 candidate variables at each split; S10 Fig):

$$\sqrt{(Predicted\ Food\ Value)}$$
$$\sim Rock\ type + Soil\ texture + pHc + pHw + Al + N\ as\ nitrate + N\ as\ ammonium + P + K$$
$$+ S + organic\ C + Fe + EC + exchangeable\ Ca + exchangeable\ Mg + exchangeable\ Na$$
$$+ exchangeable\ K + \frac{ammonium}{nitrate} + \frac{C}{N} + total\ N + SALINITY + ACIDITY$$

This initial model explained 7.88% of the variance in predicted Food Value, with a mean squared of residuals of 1.10. Nineteen of the 22 variables had a positive effect on the model, with their individual removal increasing the mean square error by up to 10% (S11 Fig). Of these, Rock type, organic carbon and ACIDITY were ranked as the most important variables. Variable selection using VSURF identified 16 variables as having a Variable Importance that exceeded a threshold level for initial inclusion in the model. Of these, ACIDITY, Rock type, exchangeable Ca and organic C were the only variables producing a positive reduction to Out-of-Bag error (S12 Fig). The final random forest model therefore included only these independent variables (with 500 trees and a random number of candidate variables at each split; S13 Fig):

$$\sqrt{(Predicted\ Food\ Value)} \sim ACIDITY + Rock\ type + exchangeable\ Ca + organic\ C$$

This final model was an improvement on the initial model, explaining 19.6% of the variance, with a mean squared of residuals of 0.96. Exclusion of each of these variables would have reduced the model's accuracy by between 6% and 17% (Fig 4A). Partial effects plots indicated that predicted Food Value was maximised when ACIDITY scores exceeded 0.5 (Fig 4B), on sedimentary rocks (Fig 4C), when exchangeable Ca was less than 18 meq 100 g$^{-1}$ (Fig 4D), and when organic C was less than 4.2%. Adverse levels of exchangeable Ca were restricted to five soil samples that had positive SALINITY scores and negative ACIDITY score (Fig 3B).

Wilcoxon-tests indicated that both ACIDITY score ($P < 0.001$, BH-adjusted $P = 0.003$; S14 Fig) and Food Value ($P < 0.0001$, BH-adjusted $P < 0.0001$), but not organic C ($P = 0.528$, BH-adjusted $P = 0.815$), were significantly higher on non-calcareous sedimentary rocks than on other rock types. t-tests indicated kernel concentrations of S ($P = 0.010$, BH-adjusted $P = 0.052$), Mg ($P = 0.021$, BH-adjusted $P = 0.077$) and P ($P = 0.036$, BH-adjusted $P = 0.109$) were also significantly higher before, but not after BH correction. None of the other elements showed a response to this rock type classification ($P > 0.05$). This included N, for which only four measurements were available from other rock types ($P = 0.815$). However, the identical median values from both rock type groupings, and the low c.v. across all rock types make it highly unlikely that a significant result would be gained even with a larger dataset for this element.

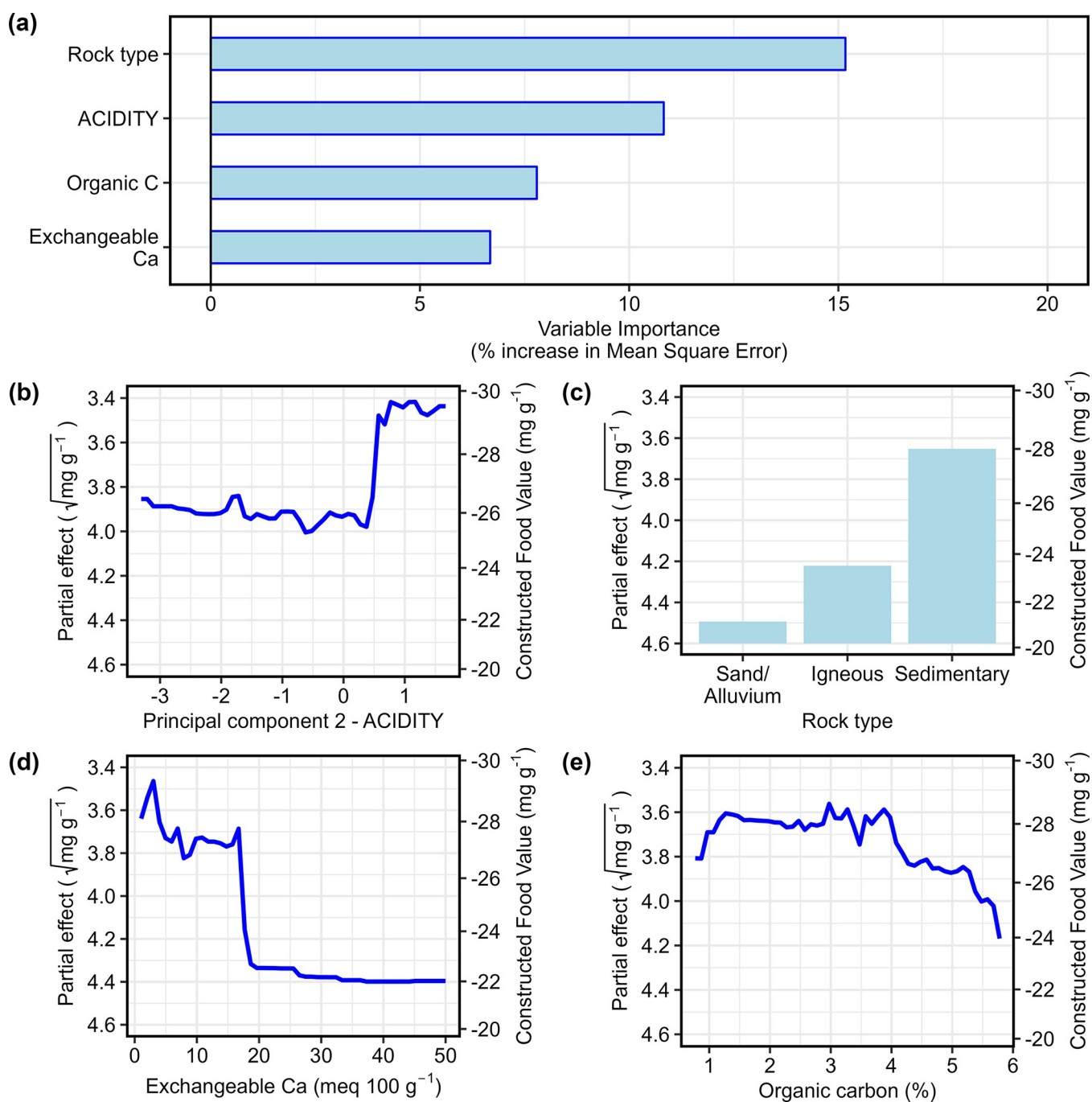

**Fig 4. Random forest model explaining the influence of geology and soils on predicted Food Value in drooping sheoak cones after variable selection using VSURF.** (a) Contribution of variables to the model. (b-e) Partial plots showing the influence of individual variables on predicted Food Value when all other variables in the model were held constant.

## Influence of rock type and species on feeding tree selection

Of the 16,922 grid cells containing black sheoak and/or forest oak in NSW, each *Allocasuarina* species was present in 52%, and 4.1% of cells contained both species. Glossy black-cockatoos were recorded feeding in these species across 5.5% of these cells, with 83.8% of feeding records being in forest oak and 16.2% in black sheoak. *Allocasuarina* species and rock type influenced

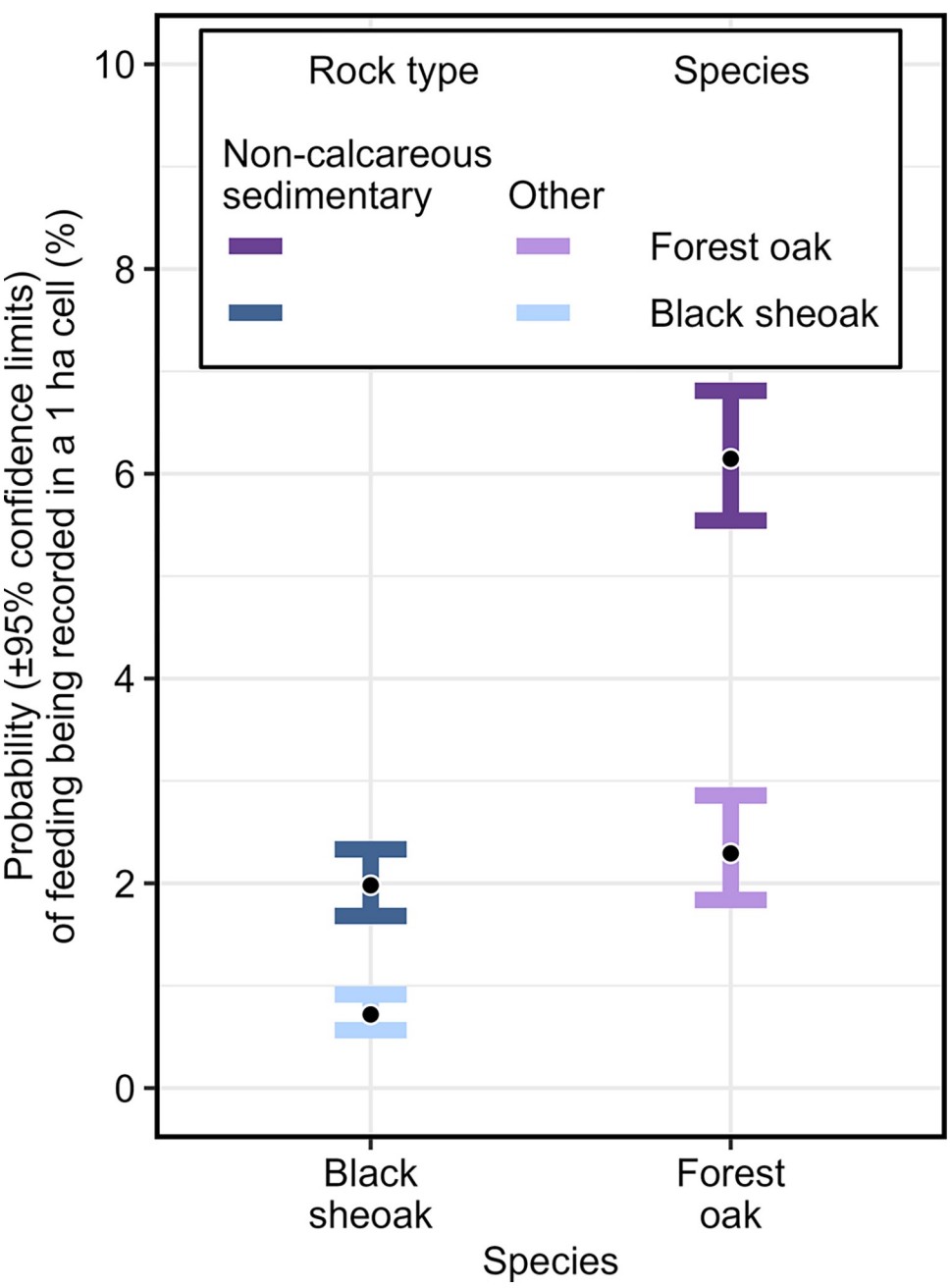

**Fig 5. Influence of species and rock type on the probability of glossy black-cockatoo feeding.** Based on a logistic generalised additive model for 16,922 one-hectare grid cells in New South Wales, after accounting for spatial variation, and with no interaction between terms.

incidence of feeding-tree selection, with significant preferences being shown for forest oak over black sheoak, and for non-calcareous sedimentary rock over other rock types (Fig 5). Both binomial generalised additive models that included or excluded interaction between species and rock type explained 11.2% of the deviance (S10 Table). The model without the interaction was chosen as the preferred model as it had the lower AIC ($\delta$AIC = 1.99) and BIC values ($\delta$BIC = 9.76); including the interaction had an insignificant effect on residual deviance ($P = 0.905$); and the interaction itself was not significant ($P = 0.908$). The preferred model

showed that–when species and location were held constant–the odds of a grid cell containing one or more cockatoo feeding records were significantly higher on non-calcareous sedimentary rocks than they were on other rock types (odds ratio: 2.78, 95% CI [2.27, 3.44], $Z = 9.67$, $P < 0.0001$). When rock type and location were held constant, the odds of feeding on forest oak were significantly higher than they were for feeding on black sheoak (odds ratio: 3.24, 95% CI [2.69, 3.90], $Z = 12.45$, $P < 0.0001$).

Ten of the geological units with at least 50 grid cells with casuarina records had a feeding: non-feeding odds ratio greater than 0.1 (S11 Table, S15 Fig). All of these units were of sedimentary rocks containing siltstone, mudstone and/or sandstone and were located along the state's north coast between Newcastle and Grafton (Fig 1A and 1B). Nine of the ten units were classified as non-calcareous sediments, two of which included minor to rare calcareous elements. Brooklana beds contained the most feeding records (275) and had the highest feeding: non-feeding odds ratio (1.4); i.e. the majority of grid cells containing casuarinas on this unit also contained feeding records. One unit (Wootton beds) had been classified as a calcareous sedimentary rock based on the presence of oolitic lime in its upper parts. This unit only contained 11 feeding records, but had a feeding:non-feeding odds ratio of 0.21. On all ten units, the majority of feeding records were in forest oak, and the odds of being a feeding tree was higher for forest oaks than it was for black sheoaks.

## Discussion

The studies described in this article demonstrated that (1) glossy black-cockatoo tree selection is based on maximising dry matter intake rate as indicated by Food Value; (2) Food Value is highest on non-calcareous sedimentary rocks; and (3) glossy black-cockatoos preferentially feed on casuarinas growing on non-calcareous sedimentary rocks.

### Tree selection

To understand how rock type affects cockatoo feeding locations, it is first essential to know how the cockatoos select their feeding trees. The results of this study confirm that selection of feeding trees by glossy black-cockatoos is based on maximising dry matter intake (i.e. Food Value) and not on the need to seek out any particular nutrient. No differences were found in the concentrations of any element between feeding and non-feeding trees, either before or after correction for false discovery error rate. The cockatoos were particularly insensitive to the nutrients most likely to be limiting–N for protein requirements and Ca for egg-production. N is considered to be the dietary nutrient most likely to be limiting to granivores [90, 91]. It is required for the production of amino acids, proteins, and other essential compounds, including DNA, RNA, vitamins and hormones [91]. Parrots require a dietary intake of 11–20% protein [92], which is essential for the production and maintenance of muscles and feathers [91, 92]. The protein content of seeds can vary between 3% and 48% [93], so active selection of seeds based on N content could be expected. However, as the protein concentration of drooping sheoak kernels (44.9 ± 0.4%) is far higher than that required by parrots, with little variation, glossy black-cockatoos do not need to seek out the most N-rich kernels. This should also be the case for cockatoos feeding on other casuarinas, with the whole seed of black sheoak containing 27.1 ± 3.1% protein [39] and those of slaty sheoak (*A. muelleriana*) containing 31.6% protein [94] (with the levels in the kernels expected to be much higher), or for Amazonian parrots feeding on the whole seed of a range of species whose protein content averaged 21.9% [averaged across 29 seed species and 17 parrot species; 95].

Ca is required for bone production and maintenance, and egg production [96]; and geologically-derived Ca has been found to limit abundance and productivity of passerines [8]. Kernel

Ca concentrations of 0.3% are at the lower end of the Ca requirements of cockatiels and budgerigars [92]. Unlike several other parrot species, glossy black-cockatoos do not appear to supplement their Ca intake with grit or invertebrates [31], so are entirely dependent on kernels to meet their Ca requirements. Low Ca intake does not appear to restrict their lifespan, which reaches a maximum of about 38 years in captivity [97], and potentially in the wild [98]. However, it may restrict their productivity, as their clutch is always a single egg [32], in comparison to most other cockatoos, which lay between two and five eggs [99]. The lack of difference in the kernel Ca concentration between feeding and non-feeding trees indicates that the birds are adapted to this low level in their diet, and so have no need to select on this variable, even though a c.v. of 25% means it should be possible to do so.

As no nutrient was found to be higher in the kernels of feeding trees than in those of non-feeding trees, the conclusion by North *et al.* (40) that the cockatoos select feeding trees based on specific nutrients is not supported. Their study analysed whole seed, which the cockatoos do not eat. The current study shows that any nutrient differences between the whole seed of feeding and non-feeding trees are simply a product of the differences in the composition of the kernels and the samara. As feeding trees have more kernel matter per cone (higher Food Value) than those of non-feeding trees, their whole seed is likely to have elevated levels of nutrients that are concentrated in the kernels (P, N, S, Mg, K and Mn, this study; fatty acids [31]), and depressed levels of nutrients that are concentrated in the samara (Ca, Na, Fe and Cu, this study; ash, fibre and carbohydrates [31]). Had the current study considered the composition of whole seed alone, as was done by North *et al.* [40], it might have erroneously concluded that the cockatoos were selecting feeding trees to minimise the intake of Cu and Zn, or to maximise N and Mn intake. Hence, I caution against making conclusions about dietary preferences of species that only consume the kernels based on whole seed.

Tree selection based on dry matter intake was originally proposed in a study of glossy black-cockatoos feeding on black sheoak [39], which found a high correlation (R = 0.607) between feeding signs and Clout's Index [an approximation of Food Value; 31]. It was later confirmed in a precursor to the current study [31]. As no other foods are taken, the casuarina kernels clearly provide all the bird's nutrient requirements, so the cockatoos select trees to maximise their intake rate alone. They achieve this by selecting trees with high Food Value, as well as trees and patches with a high density of cones [30, 100]. As the cockatoos select feeding trees on Food Value, any influence of geology on Food Value should also influence the distribution of the cockatoos.

## Determinants of Food Value

The high N and P concentrations in casuarina kernels mean that kernel production is most likely to be limited by these nutrients. Soils under drooping sheoak sampled in this study were highly infertile, with N concentrations two orders of magnitude lower than–and P concentrations less than half–that found in most Australian soils [87, 89]. However, neither soil N (including its ratio with C) nor P influenced Food Value in this study. Nor did the addition of NPK fertiliser increase either Seed Fill or Kernel Ratio in drooping sheoak on Kangaroo Island [58]. In both cases, the lack of effect likely demonstrates the effective mechanisms casuarinas have for assimilating nutrients, with cluster roots and arbuscular mycorrhizal fungi trapping P and other nutrients from the soil, and *Frankia* feeding N fixed from the atmosphere directly into the plant [6]. If Food Value in drooping sheoak were P-limited, it should be highest under alkaline conditions that are conducive to the development of cluster roots [101] and arbuscular mycorrhizal fungi [102]. The opposite was true.

If Food Value in drooping sheoak were N-limited, it should be highest under conditions conducive to nodulation and N-fixation by *Frankia*. Different strains of *Frankia* have different

pH tolerances [103]. The only casuarina strain tested was most productive at a pH of 6.5 [104]; lowering pH down to 4.5 increased nodulation in red alder (*Alnus rubra*) [105]; and high levels of $Al^{3+}$ promoted survival and growth of six casuarina strains at low pH [106]. Such responses are consistent with Food Value being maximised under a high ACIDITY score (which was negatively correlated with soil pH, and positively correlated to soil Al and Fe). It therefore seems likely that Food Value is determined by the soil conditions most suitable for *Frankia* nodulation and N-fixation, and that such conditions exist on non-calcareous sedimentary rocks, which had both significantly higher Food Value and ACIDITY scores than were found on other rock types. The reason for the negative relationship between Food Value and organic C is less clear, although some mycorrhizal fungi are negatively correlated with soil organic matter content [102].

The random forest model explained 19.6% of the variance in predicted Food Value. Other potential contributing factors include genetics, age of individual trees and the stand (including time since last fire), age of soils, disease, fungal or insect attack and rainfall [6, 55, 107, 108]. Food Value may also be affected by other factors that influence N-fixation, including soil temperature and water potential [104]. Food Value is unlikely to be limited by pollination success, as casuarinas produce large amounts of wind-dispersed pollen, and artificial pollination of drooping sheoak on Kangaroo Island increased neither Seed Fill nor Kernel Ratio [58].

### Bottom-up regulation

The preceding discussion indicates that, within the range of drooping sheoak, distribution of glossy black-cockatoos will be limited by Food Value and that this is controlled by N-fixation, which appears to be most efficient on non-calcareous sedimentary rocks. In Australia, alkaline soils are predominantly associated with carbonate lithologies (e.g. limestone, marble) and areas with a high evaporation to precipitation ratio [109]. Geological mapping identifies where sedimentary rock includes carbonate lithologies [60]. So it has been possible to test whether the relationship between Food Value and non-calcareous sedimentary rock demonstrated in drooping sheoak extends to other *Allocasuarina* species. This was indeed the case in eastern NSW, where forest oak and black sheoak were nearly three times more likely to be fed on by the cockatoos if they were growing on non-calcareous sedimentary rocks than if they were growing on other rock types. This demonstrates that geology–through its influence on nutrient pathways, particularly soil acidity–has an underpinning role in controlling the glossy black-cockatoo's distribution.

Other studies have found that soil acidity influences bird distributions, but this relationship has largely been negative. For example, abundance and/or productivity of red grouse, oven-bird, song thrush (*Turdus philomelos*) and nuthatch (*Sitta europaea*) were highest on alkaline soils [7, 8, 11]. In the case of grouse, this was linked to the abundance and nutritional status of their principal food sources [7, 11]. For ovenbirds, it appeared to be linked to the abundance of snails, their main source of Ca [8]. Ca availability has also been invoked to explain associations of other species with alkaline soils [18], but without linking evidence. No mechanism has been proposed for the association of other species to acid soils, other than to note vegetational patterns [9, 18]. Similarly, while associations of several arboreal marsupials with high fertility soils have been attributed to elevated N content in the leaves of food plants [12, 13, 16], no explanation has been proposed for the association of ground-dwelling marsupials with low fertility soils [19, 21]. For both acid soils and low fertility sites, mycorrhizal fungi that thrive in acidic, low-nutrient environments could be involved [110]. Indeed mycorrhizal fungi are important foods for many ground-dwelling mammals, including the potoroos (*Potorous* spp.) and bandicoots (*Perameles* spp.) found to be associated with low fertility soils [111].

The findings of this study have implications for any species that selects between seed-producing plants based on profitability alone. As granivores are highly sensitive to changes in profitability [27, 112, 113], they are likely to be sensitive to any aspect of soil chemistry that affects profitability. It has been demonstrated that granivores are sensitive to gross changes in abundance of food plants and seed crops, and that these changes are themselves controlled by abiotic factors, such as rainfall and fire [24, 114, 115]. However, the additional influence of abiotic factors that affect kernel production can be far from evident as kernel abundance is rarely assessed [31]. Any factor that affects profitability has the potential to affect breeding success and longevity [116, 117]. Hence, geology, through its influence on the delivery of nutrients from the soil to the kernel, has the potential to affect both granivore distributions and population dynamics.

Distributional patterns associated with particular geological units may also arise from related geomorphic features that provide habitat niches [118, 119], including of predators [120], or barriers to dispersal [121]. For example, restriction of Delacour's langur (*Trachypithecus delacouri*) to karst landscapes was found to have little to do with diet, but rather be a product of the refuge from human impacts these landscapes provide [122].

## Management implications

On the Australian mainland, glossy black-cockatoos are mainly threatened by loss of feeding habitat [123], so it is important to understand the bottom-up regulation of the population in the remnant habitat. On Kangaroo Island, from agricultural development until the mid-1990s, the species was top-down-regulated through predation by the native common brushtail possum, which had increased in number in response to vegetation clearance [32], and the birds were using only a fraction of the available drooping sheoak habitat on the island [124]. However, management of possum predation has allowed the cockatoo's population to at least double over the last two decades [38, 125]. Continued recovery could see a switch back to bottom-up regulation once the carrying capacity of the island's feeding habitat is reached, and food availability again becomes limiting. The population size at which this happens will depend on the availability of drooping sheoak trees with kernel characteristics that are adequate to support cockatoo foraging [31, 56]. Extensive bushfires (which destroyed 54% of the island's feeding habitat in 2019/20 [126], and up to 34% on the mainland [51] and are likely to increase under climate change) are likely to accelerate this shift. Hence, conservation of the species will be most effective if recovery efforts are focused in settings that have the highest carrying capacity, namely non-calcareous sedimentary rocks that support trees with high Food Value.

## Supporting information

**S1 Table. Conservation status of the glossy black-cockatoo.** Status correct as at November 20, 2023.
(PDF)

**S2 Table. Correlation matrix for Seed Fill and Kernel Ratio.** r values are displayed above the diagonal. P values are displayed below the diagonal.
(PDF)

**S3 Table. Robust regression models explaining the influence of Seed Fill and Kernel Ratio on Food Value.** FV = Food value; SF = Seed Fill; KR = Kernel ratio. Robust regression modelling was undertaken using the lmrob function in the R package robustbase.
(PDF)

**S4 Table. Processing of black sheoak and forest oak records and glossy black-cockatoo feeding records.**
(PDF)

**S5 Table. Classification of New South Wales rock types into sedimentary rocks, calcareous sedimentary rocks and other rock types.** Classification based on rock unit descriptions from Raymond OL, Liu S, Gallagher R, Zhang W, Highet LM. Surface Geology of Australia 1:1 Million Scale Dataset. 2012 Edition. Canberra: Geoscience Australia; 2012.
(PDF)

**S6 Table. Logistic model explaining the influence of predicted Food Value on the incidence of a drooping sheoak being selected as a feeding tree.** Logistic modelling was undertaken using the glm function in the R package stats.
(PDF)

**S7 Table. Elemental composition of drooping sheoak kernels.**
(PDF)

**S8 Table. Loadings for principal components analysis for soils from under drooping sheoak.** Numbers in bold indicate the dominant variables for each component. PCA was undertaken using the R package psych using a covariance matrix and oblimin rotation.
(PDF)

**S9 Table. Vectors for calculated soil variables and seed variables fit to principal component analysis for soils from under drooping sheoak.** Vector fitting undertaken using the envfit function from the R package vegan. * Vector scores adjusted for $R^2$.
(PDF)

**S10 Table. Generalized additive models explaining incidence of feeding being recorded in 1-ha grid cell in relation to tree species and rock type, after accounting for spatial variation.** Generalized additive modelling was undertaken using the R package mgcv.
(PDF)

**S11 Table. Geological units with at least 50 grid cells containing forest oak and/or black sheoak, and with feeding odds greater than 0.1.** Bold text in feeding odds field highlights feeding odds greater than 0.10. Bold text in geological description field highlights calcareous rock components. Sedimentary rocks with minor or rare calcareous components were classified as non-calcareous.
(PDF)

**S1 Fig. Fitted versus actual values in the robust regression relationship Food Value ~ Seed Fill + log(Kernel Ratio).** FV = Food value; SF = Seed Fill; KR = Kernel ratio. Robust regression modelling was undertaken using the lmrob function in in the R package robustbase. The influence of cases with large residuals was down-weighted based on a bi-square redescending score function.
(TIF)

**S2 Fig. Transformation and scaling of seed and soil nutrients.** pHc and pHw were only scaled, Al was fourth-root-transformed and scaled, all other variables were log transformed and scaled. Abbreviations: pHc, pH ($CaCl_2$ extraction); pH (water extraction); Al, aluminium; Amm.N, nitrogen as ammonium; Fe, iron; K, potassium; ex.K; exchangeable potassium; ex.Ca, exchangeable calcium; P, phosphorus; OrgC, organic carbon; ex.Mg, exchangeable magnesium; Nitr.N, N as nitrate; ex.Na, exchangeable sodium; S, sulphur; EC, electrical conductivity.

Plots were drawn using boxplot in the R graphics package.
(TIF)

**S3 Fig. Correlations between soil nutrient measures under drooping sheoak, after transformation and scaling.** Correlation coefficients shown if $P > 0.05$. Correlogram was drawn using corrplot from the R package corrplot.
(TIF)

**S4 Fig. Distribution of Food Value of drooping sheoak cones used in random forest model.** (a) original data, and (b) square-root transformed data. Plot was generated using the R package ggplot2.
(TIF)

**S5 Fig. Comparison of the elemental composition of drooping sheoak kernels with the whole seed batches from which they were extracted.** Each kernel sample was paired with the whole seed batch from which they were extracted. Sample size for nitrogen was 43, and for all other nutrients was 45. Paired t-tests were used where assumptions of normality and extreme outliers were met. Otherwise, pair-wise Wilcoxon tests were used. Plots are annotated with the relevant test statistics, P values, and the Benjamini-Hochberg-adjusted P values (adj.P). Sample sizes are in brackets. Plots were drawn using the R package ggplot2.
(TIF)

**S6 Fig. Comparison of the elemental composition of drooping sheoak kernels from non-feeding and feeding trees.** t-tests were used where assumptions of normality, equal variance and extreme outliers were met. Otherwise, Wilcoxon tests were used. Plots are annotated with the relevant test statistics, P values, and the Benjamini-Hochberg-adjusted P values (adj.P). Sample sizes are in brackets. Plots were drawn using the R package ggplot2.
(TIF)

**S7 Fig. Comparison of the elemental composition of whole seed batches of drooping sheoak from non-feeding and feeding trees.** t-tests were used where assumptions of normality, equal variance and extreme outliers were met. Otherwise, Wilcoxon tests were used. Plots are annotated with the relevant test statistics, P values, and the Benjamini-Hochberg-adjusted P values (adj.P). Sample sizes are in brackets. Plots were drawn using the R package ggplot2.
(TIF)

**S8 Fig. Summary of characteristics of soils collected under drooping sheoak.** (a) rock type, (b) soil texture, (c) pH (CaCl2 extraction), (d) pH (water extraction), (e) aluminium, (f) nitrogen as nitrate, (g) nitrogen as ammonium, (h) phosphorus, (i) potassium, (j) sulphur, (k) organic carbon, (l) iron, (m) electrical conductivity, (n) exchangeable calcium, (o) exchangeable magnesium, (p) exchangeable sodium, (q) exchangeable potassium, (r) total nitrogen, (s) ammonium N:nitrate N, and (t) carbon:nitrogen. Sample was 84 size for all samples. Plots were drawn using the R package ggplot2.
(TIF)

**S9 Fig. Scree plot showing analysis of the optimum number of components to retain in a principal component analysis of soils from under drooping sheoak.** Plot was generated using nScree from the R package nFactors.
(TIF)

**S10 Fig. Selection of parameters for the initial random forest explaining predicted Food Value in drooping sheoak.** (a) Selection of the number of candidate splits to consider at each split ($m_{try} = 21$) based on lowest Root Mean Squared Error. (b) Selection of number of trees

(500) based on the stabilisation of Mean Squared Error. Selection of $m_{try}$ was undertaken using the train function in the R package caret. Assessment of Mean Squared Error was undertaken using the R package randomForest. Plots were generated using the R package ggplot2.
(TIF)

**S11 Fig. Variable importance of the 22 variables included the initial random forest explaining predicted Food Value in drooping sheoak.** Variable Importance was calculated using R package randomForest and plotted using the R package ggplot2.
(TIF)

**S12 Fig. Ranking and selection of rock and soil variables in the initial random forest explaining predicted Food Value in drooping sheoak.** (a) Mean Variable Importance (VI) and (b) Out of Bag (OOB) error. Solid blue line is the threshold for variable inclusion in the modelling (based on VI standard deviation exceeding a minimum value predicted by a pruned classification and regression tree fitted the standard deviation curve). Dashed blue line indicates variables excluded by this selection process. Solid red line is the threshold for variable inclusion in the final model (based on a decrease in the OOB error). Variable selection was undertaken using the R package VSURF. Plots were generated using the R package ggplot2 [127].
(TIF)

**S13 Fig. Selection of parameters for final random forest model explaining predicted Food Value in drooping sheoak.** (a) Selection of the number of candidate splits to consider at each split ($m_{try}$ = 1) based on lowest Root Mean Squared Error. (b) Selection of number of trees (500) based on stabilisation of Mean Squared Error. Selection of $m_{try}$ was undertaken using the train function in the R package caret. Assessment of Mean Squared Error was undertaken using the R package randomForest. Plots were generated using the R package ggplot2.
(TIF)

**S14 Fig. Influence of rock type on soil ACIDITY, Food Value and elemental composition of kernel of drooping sheoak.** t-tests were used where assumptions of normality, equal variance and extreme outliers were met. Otherwise, Wilcoxon tests were used. Plots are annotated with the relevant test statistics, P values, and the Benjamini-Hochberg-adjusted P values (adj. P). Sample sizes are in brackets. Plots were drawn using the R package ggplot2.
(TIF)

**S15 Fig. Highly productive geological units.** Geological units with a feeding odds of at least 0.1 for either forest sheoak or black sheoak.
(TIF)

**S1 Dataset. Data for location, rock, soil, seed and kernel characteristics for drooping sheoak trees in relation to presence of, and tree selection by, glossy black-cockatoos in southeastern Australia.**
(XLSX)

**S2 Dataset. Data for location and rock type of black sheoak and forest oak records in relation to glossy black-cockatoo feeding records in New South Wales.**
(XLSX)

## Acknowledgments

I acknowledge the First Nations Peoples of the land on which the fieldwork was conducted, particularly the Ngarrindjeri (Ramindjeri), Kaurna and Narungga Peoples, custodians of

Kangaroo Island. My thanks go to the following people and organisations who assisted this project. Terry Dennis and Peter Copley (formerly Department of Environment and Heritage) oversaw the Glossy Black-Cockatoo Recovery Program and provided administrative support. Stephen Garnett (Charles Darwin University) and Michael Schultz (Riverina Wildflowers Native Nursery) assisted with collecting cone and soil samples. CSBP, Perth analysed the soils. Antonio Belperio (formerly South Australian Department of Mines and Energy) assisted with geological interpretation and provided access to unpublished geological maps. Locational data for casuarinas and glossy black-cockatoo feeding trees was sourced from the New South Wales BioNet Atlas (licence number 1495) with the assistance of Ian Geers and Philip Gleeson. Bill Venables and Allan Burr assisted with data analysis. Stephen Garnett provided advice on early versions of the manuscript, and David Gillieson advised on the final draft.

## Author Contributions

**Conceptualization:** Gabriel M. Crowley.

**Data curation:** Gabriel M. Crowley.

**Formal analysis:** Gabriel M. Crowley.

**Investigation:** Gabriel M. Crowley.

**Methodology:** Gabriel M. Crowley.

**Writing – original draft:** Gabriel M. Crowley.

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
