## [Decision Letter · Decision Letter 0]

3 Jun 2024

PONE-D-23-42022Geology controls the distribution of a seed-eating bird: Feeding-tree selection by the glossy black-cockatoo *Calyptorhynchus lathami*PLOS ONE

Dear Dr. Crowley,

Thank you for submitting your manuscript to PLOS ONE. After careful consideration, we feel that it has merit but does not fully meet PLOS ONE’s publication criteria as it currently stands. Therefore, we invite you to submit a revised version of the manuscript that addresses the points raised during the review process.

We look forward to receiving your revised manuscript.

Kind regards,

Daniel de Paiva Silva, Ph.D.

Academic Editor

PLOS ONE

Journal Requirements:

 [Funding for sample collection in 1996-1997 was provided by the Threatened Species and Communities Unit of Environment Australia, with additional funding from the South Australian Department of Environment and Heritage and the Glossy Black Rescue Fund of the National Parks Foundation, South Australia.].  

Additional Editor Comments:

Dear Dr. Crowley,

After this first review round, both reviewers believe the manuscript deserves to be published in PLoS One. Still it needs a major review before acceptance. Please take spacial care to the issues raised by both reviewers before resubmit it to PLoS One.

Sicnerely,

Daniel Silva

Reviewers' comments:

Reviewer's Responses to Questions

**Comments to the Author**

1. Is the manuscript technically sound, and do the data support the conclusions?

Reviewer #1: Yes

Reviewer #2: Yes

2. Has the statistical analysis been performed appropriately and rigorously? 

Reviewer #1: Yes

Reviewer #2: Yes

3. Have the authors made all data underlying the findings in their manuscript fully available?

Reviewer #1: Yes

Reviewer #2: Yes

4. Is the manuscript presented in an intelligible fashion and written in standard English?

Reviewer #1: Yes

Reviewer #2: Yes

5. Review Comments to the Author

Reviewer #1: The scientific manuscript entitled “Geology controls the distribution of a seed-eating bird: Feeding-tree selection by the glossy black-cockatoo Calyptorhynchus lathami”, is interesting for addressing geological aspects that shape the presence of casuarinas and therefore the distribution of the birds. The introductory part is very well written, however it is specifically missing to describe other variables that also limit the distribution of birds, as well as to describe the process of co-evolution or adaptation of the bird in depending strictly on that food resource. The problem statement must be highlighted in a better way, that is, the problem behind it and therefore why carry out this type of study. The methodology is well described and each of the analysis carried out is addressed, as are the results, in this context the discussion is discussed in a timely manner. However, some comments to be addressed and clarified in the manuscript are described in more detail below.

Line 29-33. Although nutrients are essential, other biotic and abiotic variables are important and also contribute to the formation of rocks. How this type of geology originates historically and what time scale occurs in them to form.

L.56-59. What is the structure of the seeds and will there be a mutualistic relationship between food and bird?.

L 60-62. Address the area of distribution of the species, how environmental factors influence the presence and distribution of the species.

L 87-88. Add the compass rose to the figure, that is, the orientation.

L 89-92. The information is very extensive, we suggest synthesizing more.

L 98. It is not clear, is it a bush or a tree? In what geological situations is one and the other supported.

L 105-110. By presenting these characteristics in Casuarinas, as is the effect for other plant species that grow in the soil.

L 111-112. Briefly describe the characteristics of the species.

L 118. What historical evolutionary process or other factor caused the species to depend on this food resource.

L 134-136. This is confusing. It is indicated that there is a study as background that provides the opportunity to evaluate the variables in seeds and geology. Clarify this wording.

L 144-153. Argue if the samples used are sufficient, because, based on whom and what the bias would be or not.

L 168-170. What is the bias or not in selecting the samples in that way, argue.

L 235-241. What software was used to make these projections?

L 260-261. Support why this model was selected as residual.

L 279-283. Synthesizing the information is very extensive.

L 342-350. Synthesize the information, be more specific.

L 377-383. Synthesize the information.

L 464-471. Because it is assumed that this would have been achieved, sustain.

Reviewer #2: the article was very well thought out and written, I congratulate the authors for their effort in carrying out the analyzes and creating a well-done text.

I only have a few suggestions:

Line 45: "in several species in Australia [9, 10] and Africa [11] and North America [12]." add "," ; "in several species in Australia [9, 10], Africa [11] and North America [12]."

Line 436: here you could explain more why nitrogen influences animal abundance.

Line 441: what are these foods preferred by Amazon parrots? what types of seeds?

6. PLOS authors have the option to publish the peer review history of their article (what does this mean?). If published, this will include your full peer review and any attached files.

Reviewer #1: No

Reviewer #2: No

---

## [Author Response · Author response to Decision Letter 0]

11 Jun 2024

Response to reviewers has been provided in the attached file.

---

## [Decision Letter · Decision Letter 1]

23 Jul 2024

Geology controls the distribution of a seed-eating bird: Feeding-tree selection by the glossy black-cockatoo *Calyptorhynchus lathami*

PONE-D-23-42022R1

Dear Dr. Crowley,

We’re pleased to inform you that your manuscript has been judged scientifically suitable for publication and will be formally accepted for publication once it meets all outstanding technical requirements.

Kind regards,

Daniel de Paiva Silva, Ph.D.

Academic Editor

PLOS ONE

Additional Editor Comments (optional):

Reviewers' comments:

Reviewer's Responses to Questions

**Comments to the Author**

1. If the authors have adequately addressed your comments raised in a previous round of review and you feel that this manuscript is now acceptable for publication, you may indicate that here to bypass the “Comments to the Author” section, enter your conflict of interest statement in the “Confidential to Editor” section, and submit your "Accept" recommendation.

Reviewer #1: All comments have been addressed

2. Is the manuscript technically sound, and do the data support the conclusions?

Reviewer #1: Yes

3. Has the statistical analysis been performed appropriately and rigorously? 

Reviewer #1: Yes

4. Have the authors made all data underlying the findings in their manuscript fully available?

Reviewer #1: Yes

5. Is the manuscript presented in an intelligible fashion and written in standard English?

Reviewer #1: Yes

6. Review Comments to the Author

Reviewer #1: (No Response)

7. PLOS authors have the option to publish the peer review history of their article (what does this mean?). If published, this will include your full peer review and any attached files.

Reviewer #1: No

---

## [Editor Report · Acceptance letter]

1 Aug 2024

PONE-D-23-42022R1 

PLOS ONE

Dear Dr. Crowley, 

I'm pleased to inform you that your manuscript has been deemed suitable for publication in PLOS ONE. Congratulations! Your manuscript is now being handed over to our production team.

Kind regards, 

on behalf of

Dr. Daniel de Paiva Silva 

Academic Editor

PLOS ONE